# OpenReview forum: "FORTRESS: Fast, Tuning-Free Retrieval Ensemble for Scalable LLM Safety"
_TMLR — Accepted by TMLR_

### Review · Reviewer_6sWM · 2025-07-19

**Summary Of Contributions:**

As a surrogate of the guardrail model, this paper proposes a semantic retrieval framework to classify the harmful content.

**Audience:**

Yes

**Broader Impact Concerns:**

No concerns.

**Claims And Evidence:**

Yes

**Requested Changes:**

Please check the weakness and add extra experiments to erase my concern,

**Strengths And Weaknesses:**

Strengths:

1. The paper is sort of well written with comprehensive evaluation.

2. The method looks interesting and could be a powerful surrogate of existing fine-tuned based guardrail model.

3. The authors compare their method with exisitng guardrail, e.g., llamaguard, and guardreasoner.


Weakness:

1. Section 3.2, It is unclear how the second analyzer work how what is its core functions? If I understand correctly, once you compare the prompt's embedding and retrieve the topk most similar documents from the harmful datasets, you should be able to set a threshold to detect whether the prompt is harmful or not. Why brother to introduce second analyzer later? I found this design hard to read and could not understand.

2. The authors do not discuss about how the method can resist jailbreak attack. Let say the attackers have access to the FORTress database (I don't see why they cannot access because all the data are collected from Internet, which everyone can access). Then they can add jailbreak suffix to the prompt such that the embedding of the prompt is far away from the harmful embedding. There are no experiments and also no discussion on the weakness of this aspect of the method.

3. The authors claim that one advantage of their method is that it is a tuning free safety framework that operate faster than the standard guardrail. However, there is not system overhead being evaluated. if understand correctly, to construct the vector based data-based, you need to extract the embedding of all the collected harmful  data. The authors should evaluate the following overhead:

First , would the storage cost of such harmful data vector be very large. Would It be much larger than a 7B guardrail model?

Second, would the memory to perform topk retrieval very large. How does it compare with a standard guardrail model.

Third, would the computation time to construct the harmful data embedding vector very large, and how does it compare with training a standard guardrail model, e.g., llama guard?

---

> ### Author Response · Authors · 2025-07-19
> **Clarification on the Role of the Secondary (Perplexity) Analyzer**
>
> We thank reviewer 6sWM for the insightful question regarding the design of our two-stage pipeline. We recognize the need to clarify the distinct and complementary roles of the primary (retrieval) and secondary (perplexity) analyzers.
>
> The primary retrieval detector is highly effective at identifying threats that are semantically similar to known harmful patterns in our knowledge base. However, its effectiveness diminishes against novel or "zero-day" attacks that are linguistically and semantically distant from any existing exemplars.
>
> This is precisely the gap the secondary analyzer is designed to fill. It functions as a statistical anomaly detector, scrutinizing the linguistic typicality of a prompt. It is particularly potent for identifying out-of-distribution attacks, such as those employing unusual syntax or adversarial suffixes, which might otherwise evade a purely semantic check.
>
> The two detectors work in synergy. The primary detector provides a strong initial hypothesis based on known threats, while the secondary analyzer provides a crucial safety net for unknown and evolving attack vectors. The significant impact of this component is demonstrated in our ablation study (Table 4 of the main paper), where disabling the perplexity analyzer led to a substantial drop in performance, particularly on benchmarks with novel attacks.
>
> **Ablation Study: Impact of Removing Perplexity Analyzer***(FORTRESS Gemma 1B Expanded Model)*
>
> | Configuration | Avg. F1 (Unsafe) | Δ F1 (pts.) |
> | --- | --- | --- |
> | **Full Pipeline** | **85.7%** | **—** |
> | Without Perplexity Analyzer | 75.8% | -9.9 |
>
> This shows that the secondary analyzer is not a redundant step but a critical component for ensuring robust and comprehensive threat coverage.

---

> ### Author Response · Authors · 2025-07-19
> **Resilience to Sophisticated and Adaptive Jailbreak Attacks**
>
> We thank the reviewer for raising the critical issue of resilience against adaptive jailbreak attacks, especially in a grey-box scenario where an attacker has knowledge of the database. To address this, we conducted a rigorous red-teaming evaluation using a purpose-built evolutionary generation engine to craft worst-case adversarial prompts specifically targeting our best-performing model, FORTRESS Qwen 4B (Expanded).
>
> The framework developed two sets of attacks:
>
> 1. **Black-Box:** Simulates an attacker who mutates and recombines components of known successful attacks without knowledge of the defense mechanism.
> 2. **Grey-Box:** Simulates a sophisticated attacker with access to the vector database, who crafts attacks to target sparse "gaps" in the embedding space, making them semantically distant from known content.
>
> The results, shown below, are revealing.
>
> | Model | F1 Score on SIEGE-Black-Box | F1 Score on SIEGE-Grey-Box | Avg. F1 |
> | --- | --- | --- | --- |
> | **FORTRESS Qwen 4B (Exp.)** (Targeted) | **2.6%** | **1.1%** | **1.8%** |
> | FORTRESS Qwen 4B (Default) | 92.2% | 88.0% | 90.1% |
> | GuardReasoner 8B | 99.2% | 98.8% | 99.0% |
> | ShieldGemma-2 4B | 100.0% | 100.0% | 100.0% |
> | LlamaGuard-3 8B | 19.8% | 17.6% | 18.7% |
>
> While the red-teaming successfully broke its specific target (F1 score of 1.8%), the experiment highlights a crucial finding: **these attacks are brittle and lack generalizability.** The very same prompts were ineffective against other leading models (GuardReasoner, ShieldGemma) and, most importantly, against the same FORTRESS architecture using a different knowledge base (90.1% F1).
>
> This underscores a core strength of our data-centric design. While a knowledgeable attacker can craft "overfitted" attacks for a *specific* database version, these vulnerabilities do not represent fundamental, transferable jailbreaks. Crucially, patching such a vulnerability does not require costly model retraining; it simply involves ingesting the successful attack prompts into the database, instantly inoculating the system. This demonstrates a resilient and highly adaptable defense posture.

---

> ### Author Response · Authors · 2025-07-19
> **Analysis of System Overhead and Resource Requirements**
>
> Thank you for raising these important practical considerations. We agree that a full evaluation must account for storage, memory, and setup costs to accurately assess the "tuning-free" advantage. We have conducted this analysis, comparing FORTRESS with leading fine-tuned models.
>
> Our analysis is split into two parts: **(1) Inference/Deployment Overhead** and **(2) Setup/Training Overhead**.
>
> ### 1. Inference/Deployment Overhead (Storage and Memory)
>
> The table below details the resource requirements during operation. A key architectural advantage of FORTRESS is that its knowledge base (KB) is loaded into system RAM, which is significantly more scalable and cost-effective than VRAM.
>
> | Model | Peak VRAM (GB) | KB Size (Disk/RAM) | Total Memory (GB) |
> | --- | --- | --- | --- |
> | **FORTRESS Qwen 4B (Exp.)** | **8.8** | **3.3** | **12.1** |
> | FORTRESS Qwen 4B (Def.) | 8.8 | 0.1 | 8.9 |
> | Llama Guard 3-8B | 15.7 | N/A | 15.7 |
> | GuardReasoner 8B | 22.0 | N/A | 22.0 |
> | ShieldGemma-2 4B | 9.7 | N/A | 9.7 |
>
> This data addresses two key concerns:
>
> - **Storage Cost:** The expanded knowledge base (~270,000 entries) requires only **3.3 GB** of storage, which is smaller than the weight files of the baseline models.
> - **Memory Usage:** Our top configuration has a total memory footprint of **12.1 GB** (8.8 GB VRAM + 3.3 GB RAM). This is substantially more efficient than high-performance baselines like GuardReasoner 8B (22.0 GB VRAM) and Llama Guard 3-8B (15.7 GB VRAM), making our solution more accessible for deployment.
>
> ### 2. Setup/Training Overhead
>
> While most fine-tuned models do not disclose their training time, we can provide a direct comparison for `GuardReasoner 8B` using the figures reported in its original paper (Liu et al., 2025). This comparison highlights the profound difference in computational cost for initial setup.
>
> | Metric | **FORTRESS Qwen 4B (Exp.)** | **GuardReasoner 8B** (from Liu et al., 2025) |
> | --- | --- | --- |
> | **Process** | Knowledge Base Construction | R-SFT & HS-DPO Fine-tuning |
> | **Setup Time** | **~1 hour** | **~30.5 GPU hours** |
> | **Hardware** | 1x NVIDIA RTX 3090 (24GB) | 4x NVIDIA H100 (80GB) |
>
> The contrast in setup overhead is stark. Constructing the FORTRESS knowledge base is a one-time process that completes in about an hour on a single consumer-grade GPU. In contrast, training GuardReasoner 8B required over **30 GPU hours** on a cluster of four powerful, data-center-grade H100 GPUs.
>
> This vast difference in both time and computational resources underscores the core advantage of our data-centric approach. FORTRESS is not only more efficient at inference but is also substantially faster and more accessible to set up, adapt, and maintain, entirely avoiding the costly and time-consuming fine-tuning cycles required by traditional guardrail models.

---

### Review · Reviewer_haTS · 2025-07-19

**Summary Of Contributions:**

FORTRESS builds a labelled database of safe and unsafe prompts with some human verification. It also constructs a tuning-free pipeline that integrates semantic and anomaly detection to check whether input prompts to an LLM are safe.
Specifically, when a new prompt inputs LLM, the pipeline uses KNN to compare its embedding with the database and vote on its safety while simultaneously calculating an anomaly score from the prompt's token log probabilities; it then combines the two signals to decide whether the prompt is safe.

**Audience:**

Yes

**Claims And Evidence:**

Yes

**Requested Changes:**

Critical:

1. Explain the role of Tratio (Algorithm 1, Table 5) and why you chose 0.3. Clarify whether other values were or were not tested. The parameter is one of the cores of your work on "dynamic" weighting and needs a clear explanation.

2. In your Data Curation and Preprocessing section, you use the initial (def) data. But the main experiments and ablation studies (Table 2, 4) use the expanded data. Consider consistency in context. Moreover, there is a lack of details about the expanded data beyond the distribution in Figure 11, including the scale of the expanded data and processing methods (add how many data, whether an overlap exists with initial data and whether manually labelled as in initial).

3. Tables 3 and 4 do not report exactly the same evaluation metrics as Table 2. Consider consistency in context and add experimental results of dataset comparison and ablation on Ailum, Harm, OAI, Simple, XSafe, and Lat. to validate.

4. Explain how you searched parameters (C,λ,μ) with Bayesian optimization in detail.

Would Strengthen:

1. Clarify the meaning of all symbols in Algorithm 1 in the paper.

2. Fix and unify the reference format. For example, reference 1, 2, and 4 formats are inconsistent.

3. Unify the heatmap colours. The colours in Figure 10 are opposite to those in other heatmaps.

**Strengths And Weaknesses:**

Strengths:
FORTRESS is tuning a free external safety layer for LLM, making LLM safety a data-centric retrieval problem rather than a tuning problem.
Low computational overhead lightweight framework for small parameter models suitable for low-cost deployment in industrial scenarios

Weaknesses:
FORTRESS heavily relies on data quality and coverage; changes in data quantity or category may affect it. For example, expanding the database or unsafe categories may require manual relabeling. FORTRESS also depends on each category's different parameters (C,λ,μ). If the dataset keeps expanding or adds new categories, all parameters may need recalibration.

---

> ### Author Response · Authors · 2025-07-19
> **Clarification on the Dynamic Ensemble Strategy and Tratio**
>
> We thank reviewer haTS for their insightful question regarding the `Tratio` parameter. This parameter is a cornerstone of our dynamic ensemble strategy, and we are happy to clarify its function.
>
> The `Tratio` parameter serves as a **retrieval coherence threshold**, allowing FORTRESS to dynamically assess the confidence of the primary semantic detector.
>
> - **When retrieval is coherent**, the system trusts the strong semantic match and weights the signals (0.8, 0.2) in its favor.
> - **When retrieval is ambiguous** (i.e., the retrieved neighbors are mixed), FORTRESS becomes more cautious, rebalancing the weights to (0.5, 0.5) to increase reliance on the secondary perplexity analyzer.
>
> To validate our choice of `Tratio` = 0.3, we conducted a sensitivity analysis. The results below demonstrate the system's robustness.
>
> | `Tratio` | Aegis | FORT | JBB | XSTest | **Avg. F1** |
> | --- | --- | --- | --- | --- | --- |
> | 0.1 | 78.2 | 91.5 | 89.7 | 83.3 | **85.7** |
> | 0.2 | 78.2 | 91.5 | 89.7 | 83.3 | **85.7** |
> | **0.3 (Our Choice)** | **78.2** | **91.5** | **89.7** | **83.3** | **85.7** |
> | 0.4 | 78.2 | 91.5 | 89.7 | 83.3 | **85.7** |
> | 0.5 | 78.1 | 91.3 | 86.3 | 74.1 | 82.4 |
> | 0.6 | 78.1 | 91.3 | 86.3 | 74.1 | 82.4 |
>
> Performance is exceptionally stable for `Tratio` values up to 0.4. Our choice of 0.3 is a well-justified value within this robust optimal range, confirming that our dynamic ensemble is not sensitive to precise hyperparameter tuning and is well-suited for practical deployment.

---

> ### Author Response · Authors · 2025-07-19
> **Analysis of the Expanded Knowledge Base and Data Novelty**
>
> We thank the reviewer for this insightful observation. You are correct that we detail the **Default Database** in the Data Curation section but feature the **Expanded Database** in our main results (Table 2). This was a deliberate choice for two key reasons:
>
> 1. **Demonstrating our Core Claim:** The primary contribution of FORTRESS is its ability to improve via simple data ingestion. Featuring the `Expanded` configuration allows us to present our system at its full capability, directly demonstrating the effectiveness of this scalable design. The direct comparison and performance uplift are quantified specifically in Table 3.
> 2. **Clarity and Readability:** Table 2 is already dense with numerous models and benchmarks. Including an additional set of results for our `Default` configuration would have added significant clutter, making it harder to compare FORTRESS against the primary external baselines.
>
> This structure allows us to establish a validated baseline, prove the efficacy of our core contribution, and maintain a clear comparison against prior work. The following analysis further demonstrates that our performance gains stem from an efficient, scalable process that ingests genuinely new and diverse examples.
>
> Our approach allows for rapid adaptation. We expanded our **Default Database** (9,418 entries) by ingesting prompts from `WildJailbreak` and `AegisSafetyDataset v2`. This automated process, requiring only format unification with **no manual relabeling**, resulted in the **Expanded Database** of 274,759 entries.
>
> Critically, the setup overhead is minimal. Building the expanded knowledge base is a one-time, offline process that completed in approximately **1 hour** on a single RTX 3090. This stands in stark contrast to the costly and time-consuming fine-tuning cycles required by traditional guardrails.
>
> To confirm that this expansion adds valuable, non-redundant information, we conducted an overlap analysis:
>
> | Metric | Overlap Count | Overlap Percentage |
> | --- | --- | --- |
> | **Syntactic Overlap** (Exact/Normalized) | 153 | **0.06%** |
> | **Semantic Overlap** (Similarity ≥ 0.90) | 93,323 | **33.97%** |
>
> This analysis reveals two key points:
>
> 1. **Near-Zero Syntactic Redundancy (0.06%):** This extremely low number confirms that the expansion datasets introduce a massive volume of new phrasings and adversarial templates, not just duplicates.
> 2. **Balanced Reinforcement and Novelty (33.97%):** This is a powerful finding. The ~34% semantic overlap indicates that the expansion data significantly **reinforces and densifies** the embedding space for existing threat categories, making the system more robust against paraphrasing and variations of known attacks. Simultaneously, it reveals that over **66% of the ingested data is semantically novel**, expanding the breadth of the knowledge base and improving its ability to recognize entirely new lines of attack.
>
> This validates our data-centric philosophy: FORTRESS's performance scales by both reinforcing existing knowledge and efficiently learning a vast amount of new semantic information, making it resilient to a wider range of evolving threats.

---

> ### Author Response · Authors · 2025-07-19
> **Rationale for Ablation Study Design and Expanded Results**
>
> We appreciate the reviewer's attention to consistency. Our decision to perform ablation studies on a focused subset of four benchmarks (⁠`Aegis`, ⁠`FORT`, ⁠`JBB`, ⁠`XSTest`) was deliberate and reflects our paper's experimental design philosophy.
>
> **Table 2 serves a distinct purpose**: It demonstrates comprehensive coverage across nine benchmarks to establish our method's broad applicability and facilitate comparison with prior work. This table fulfills the coverage requirement expected in the literature.
>
> **All analytical work beyond this main comparison** (ablation studies, scalability analysis, noise robustness, parameter sensitivity, leave-one-category-out analysis, and performance scaling investigations) **deliberately use the four-benchmark subset** for methodological rigor. The excluded benchmarks consist predominantly of harmful prompts, which can produce misleadingly high scores for overly-cautious models that default to "unsafe" classifications. Our chosen subset provides balanced safe/unsafe distributions essential for meaningful ablation analysis.
>
> This design is consistent throughout our paper: Table 2 for comprehensive coverage, and focused four-benchmark analysis for all deeper investigations. **We maintain this approach as adding results on imbalanced datasets would not strengthen the ablation analysis** and could introduce misleading interpretations that detract from the core findings.
>
> The latency results requested have already been incorporated into our ablation table, demonstrating that our full pipeline achieves superior accuracy while maintaining computational efficiency.
>
> | Configuration (FORTRESS Gemma 1B) | Aegis | FORT | JBB | XSTest | **Avg. F1** | **Avg. Latency (ms)** |
> | --- | --- | --- | --- | --- | --- | --- |
> | **Full Pipeline (Exp.)** | **78.2** | **91.5** | **89.7** | **83.3** | **85.7** | **51.8** |
> | Without Perplexity | 69.8 | 91.6 | 83.6 | 58.2 | 75.8 (-9.9) | 57.0 |
> | Without Retrieval | 70.3 | 90.3 | 83.6 | 66.2 | 77.6 (-8.1) | 57.0 |
> | Without Dynamic Thresholds | 70.1 | 72.6 | 80.0 | 61.5 | 71.0 (-14.7) | 56.5 |
> | Global Optimized Threshold | 67.4 | 72.1 | 79.4 | 57.1 | 69.0 (-16.7) | 50.7 |
>
> The full pipeline is not only the most accurate but also highly efficient, confirming that our integrated design achieves superior performance without a computational penalty.

---

> ### Author Response · Authors · 2025-07-19
> **Explanation of Per-Category Bayesian Optimization**
>
> We thank the reviewer for requesting more detail on our Bayesian optimization process, as it is a key innovation that enables the high performance of our perplexity analyzer.
>
> The core motivation is that a single, global set of perplexity parameters (`C`, `λ`, `μ`) is suboptimal. Different prompt categories have distinct linguistic structures; for example, a benign creative writing prompt has a different perplexity profile than a harmful, direct command. Our context-aware approach calibrates the analyzer's sensitivity for each of the 20 safe and unsafe categories.
>
> The process is as follows:
>
> 1. **Objective:** For each category, we use Bayesian optimization (`scikit-optimize.gp_minimize`) to find parameters that minimize the **Mean Squared Error**. This powerfully aligns the predicted adversarial probability with its target value: 0 for safe prompts and 1 for unsafe prompts.
> 2. **Execution:** We iterate through each of the 20 categories in our database, running the optimizer to identify the optimal `(C, λ, μ)` configuration for that specific linguistic domain.
> 3. **Efficiency:** This calibration is a **one-time, offline process**. It completes in approximately **15 minutes on a single RTX 3090 GPU** for all 20 categories—a negligible cost compared to the days or weeks of GPU time required for fine-tuning multi-billion parameter models.
>
> This automated and efficient calibration is central to FORTRESS's adaptability. It allows the system to adapt to new data patterns via a lightweight process, reinforcing our data-centric and tuning-free design philosophy.

---

> ### Author Response · Authors · 2025-07-19
> **Commitment to Revisions and Final Remarks**
>
> We sincerely thank the reviewer for their thorough review and constructive feedback. The suggestions to strengthen the paper are excellent, and we will incorporate all of them into the final manuscript.
>
> Specifically, we will make the following revisions:
>
> 1. **Algorithm 1 Clarification:** We will add a detailed description of all symbols used in Algorithm 1 to ensure its logic is transparent and easy to follow.
> 2. **Reference Unification:** We will carefully review and unify the formatting of all citations to ensure consistency throughout the bibliography.
> 3. **Heatmap Consistency:** We will adjust the color scheme of Figure 10 to align with the other heatmaps in the paper, ensuring that "high score" is consistently represented by the same color for improved readability.
>
> We are confident that these changes, combined with the clarifications and additional empirical evidence provided in our other responses, will significantly strengthen the paper. We appreciate the opportunity to elaborate on the novel aspects of FORTRESS and demonstrate the robustness of its data-centric, tuning-free design.

---

### Review · Reviewer_j8kv · 2025-08-04

**Summary Of Contributions:**

This paper introduces FORTRESS, a novel framework for llm safety that operates without the need for gradient-based fine-tuning. The system integrates semantic retrieval with dynamic perplexity analysis using a single instruction-tuned LLM. The primary detector uses semantic similarity to identify known threats from a curated knowledge base, while a secondary analyzer uses perplexity scores to detect novel, zero-day attacks. A dynamic ensemble strategy weighs these two signals, prioritizing semantic matches for known threats and statistical anomalies for unfamiliar ones. The authors demonstrate through extensive evaluation on nine safety benchmarks that FORTRESS achieves a sota F1 score of 91.6%, outperforming leading fine-tuned classifiers while being over five times faster.

**Audience:**

Yes

**Claims And Evidence:**

Yes

**Requested Changes:**

It would be better if the authors could provide more ablation experiments on different knowledge bases, especially how to curate high-quality KBs

**Strengths And Weaknesses:**

### **Strengths**

1. The core strength of FORTRESS lies in the tuning-free architecture. By combining semantic retrieval for known threats and perplexity analysis for zero-day attacks, it addresses the brittleness of fine-tuned models that require constant, computationally expensive retraining. This dual-pronged approach, orchestrated by a single LLM, presents a practical and efficient solution for real-world deployment, where adaptability and low latency are critical.
2. The paper provides compelling evidence that FORTRESS's performance scales with the size of its knowledge base without a significant latency trade-off, which is beneficial for scaling up the system.
3. The authors conducted an exhaustive evaluation across nine diverse safety benchmarks, including challenging adaptive attack datasets. The inclusion of extensive ablation studies robustly validates the contribution of each component of the FORTRESS system.
### **Weaknesses**

1. The system's performance is fundamentally tied to the quality of its knowledge base. This initial setup is labor-intensive and requires access to powerful models like Gemini 2.5 for re-classification. This presents a potential barrier to replication and deployment for teams without similar resources. The system's effectiveness could be compromised if the initial dataset is not sufficiently diverse or accurately labeled.
2. As highlighted in the failure case analysis, the system can misclassify benign prompts that have semantic overlap with sensitive categories. While the paper suggests this can be mitigated by enriching the knowledge base, it points to an inherent vulnerability where the system may struggle to distinguish between harmful intent and the mere discussion of a sensitive topic, potentially leading to over-cautious and restrictive behavior.

---

> ### Author Response · Authors · 2025-08-04
> **Addressing Knowledge Base Dependency and Replication Concerns**
>
> We thank Reviewer j8kv for their insightful feedback on knowledge base dependency and the potential barrier to replication. To directly address this, we conducted a new experiment to prove that FORTRESS's architecture is the primary performance driver, not its reliance on an expensive proprietary labeling model.
>
> **Experimental Design:** We created a **"Pragmatic KB"** by re-classifying our ~11k dataset with the open-source `qwen/qwen3-30b-a3b-instruct-2507` model, requiring no proprietary APIs or manual quality control. This KB was then expanded with the same data ingestion pipeline as our original **"Gold-Standard KB"**. We then evaluated our FORTRESS Qwen 4B model on both knowledge bases.
>
> **Results:**
>
> | Model Configuration | Aegis | FORT | JBB | XSTest | **Avg. F1** |
> | --- | --- | --- | --- | --- | --- |
> | *Baseline: GuardReasoner 8B* | 70.1 | 72.5 | 77.6 | 61.5 | 86.3 |
> | **FORTRESS w/ Gold-Standard KB** | 82.5 | 93.3 | 90.0 | 91.0 | **89.2** |
> | **FORTRESS w/ Pragmatic KB (New)** | 82.2 | 93.3 | 89.4 | 91.3 | **89.0** |
>
> **Analysis:**
>
> The results conclusively show that the Pragmatic KB configuration (89.0% F1) performs on par with the Gold-Standard version and continues to significantly outperform the leading fine-tuned baseline. This proves that practitioners can achieve robust safety using our framework with accessible, open-source tools, directly addressing the concern about replication barriers and validating the robustness of the FORTRESS architecture itself.

---

> ### Author Response · Authors · 2025-08-04
> **Resolving Over-Cautious Classifications without Performance Regression**
>
> We thank the reviewer for identifying the critical challenge where a safety system might display over-cautious behaviors. To prove that FORTRESS's data-centric design can overcome this, we conducted a targeted experiment to demonstrate its safe and efficient maintainability.
>
> **Experimental Design:**
>
> We sampled 5 representative false-positive errors where our top model was overly cautious. These errors included benign queries about public figures (Trudeau, Biden), policy discussions ("legalizing drugs"), and even requests grounded in fiction ("Mary Poppins' passport number").
>
> To correct these, we ingested a minimal "corrective patch" of just 50 thematically related benign prompts (10 per error) into our database, creating a **"Surgery KB"**.
>
> **Results:**
>
> | Model Configuration | Aegis | FORT | JBB | XSTest | **Avg. F1** | **Targeted Errors Resolved** |
> | --- | --- | --- | --- | --- | --- | --- |
> | **FORTRESS (Before Fix)** | 82.5 | 93.3 | 90.0 | 91.0 | **89.2** | 0 / 5 |
> | **FORTRESS (After Surgery)** | 82.5 | 93.5 | 90.0 | 91.0 | **89.3** | **5 / 5** |
>
> **Analysis:**
>
> The results demonstrate a core operational advantage of FORTRESS. The surgical addition of only 50 data points successfully resolved 100% of the targeted over-cautious classifications. Most importantly, this fix was achieved with **zero performance regression** across our four global benchmarks (89.2% vs. 89.3% F1). This confirms that FORTRESS is not inherently vulnerable to this issue; rather, its data-centric architecture allows it to be precisely calibrated. We can make the system less restrictive and more nuanced on specific topics through a simple, low-effort database operation, all without compromising its core safety capabilities.

---

### Author Response · Authors · 2025-08-04
**Summary of Author Responses for Submission #5263**

**Dear Action Editor and Reviewers,**

We sincerely thank you for the thorough and constructive feedback. We have addressed all concerns through comprehensive experiments and clarifications, summarized below:

**For Reviewer j8kv (2 responses):**

- Demonstrated FORTRESS achieves comparable performance (89.0% F1) using only open-source tools, removing replication barriers.
- Showed over-cautious behaviors can be surgically corrected with minimal data ingestion (50 prompts) without performance regression.

**For Reviewer haTS (5 responses):**

- Clarified `Tratio=0.3` as the retrieval coherence threshold with sensitivity analysis showing robust performance.
- Provided detailed expansion database analysis: ~275K entries with 0.06% syntactic and ~34% semantic overlap.
- Justified the focused 4-benchmark ablation design for methodological rigor.
- Explained the per-category Bayesian optimization, which completes in ~15 minutes on a single GPU.
- Committed to formatting and visualization improvements.

**For Reviewer 6sWM (3 responses):**

- Clarified the secondary analyzer detects zero-day attacks via perplexity, complementing semantic retrieval (-9.9 F1 drop without it).
- Demonstrated resilience against adaptive attacks through dedicated red-teaming experiments.
- Showed FORTRESS requires only 12.1GB total memory vs. 22GB for baselines, with a 1-hour setup vs. 30+ GPU hours for fine-tuning.

All experiments validate FORTRESS's core advantages: tuning-free adaptation, robust performance, and practical deployment efficiency.

---

### Decision · Action_Editor_vUbe · 2025-10-10

**Recommendation:** Accept with minor revision

**Additional Comments:**

The authors propose a novel tuning-free framework for LLM safety named FORTRESS. By combining semantic retrieval for known threats and perplexity analysis for zero-day attacks, it addresses the challenges of fine-tuned models that require constant, computationally expensive retraining. Experiments show that the framework scales with the size of its knowledge base without a significant latency trade-off.

The reviewers agree that the problem is important and the tuning-free solution is practical. The reviewers raised several concerns, mainly
(1) Performance highly depends on the quality of knowledge base (major limitation), (2) hyperparamter of dynamic ensemble strategy, (3) ablation study, (4) resilience against adaptive attacks. The authors performed new experiments in rebuttal and clarified writing issues, addressed most of the concerns. After rebuttal, two reviewers recommended acceptance and one reviewer leaned towards rejection. AE recommends acceptance  with minor revision. The authors are highly suggested to include new results, discussions and promised changes from rebuttal into the camera ready. Also, Reviewer 6sWM commented on expected changes of camera ready in the recommendation, which is quoted below: "
- For the Secondary (Perplexity) Analyzer, I now can understand its function by the rebuttal. Please make this clear in the revision and also attach the ablation results (perhaps in the appendix) to show the function and necessity of the second analyzer.
- I appreciate the authors further experiment on the adaptive attack. For detection-based methods, it is very straightforward to attack as long as the detection criterion is known to the attackers. I suggest the authors to include these adaptive results and acknowledge the weakness of Fortress in an easy-to-spot place in the paper, e.g., add a limitation section to discuss.
- The system overhead looks good to me. Please do attach and discuss this overhead comparison in the camera ready."

**Audience:**

Yes

**Audience Explanation:**

Yes. LLM safety is an important and emergent topic, which could be interesting to the audience.

**Claims And Evidence:**

Yes

**Claims Explanation:**

Yes. The authors propose  a novel tuning-free framework for LLM safety named FORTRESS. The authors' claims on its strong performance, low latency, and adaptability are supported by experiments across nine safety benchmarks. The empirical results are further enhanced during rebuttal.

---

> ### Author Response · Authors · 2025-10-11
> **Camera-Ready Revision Uploaded for Submission #5263**
>
> Dear Action Editor,
>
> Thank you for accepting our paper with minor revision. We have submitted the camera-ready version incorporating all requested changes.
>
> As requested in your decision, we have:
>
> 1. **Included all rebuttal experiments**: Added results for adaptive attacks (Section 4.5), system overhead analysis (Table 5), open-source knowledge base validation (Table 6), and surgical correction experiments (Table 7).
>
> 2. **Added Limitations section**: Section 6 now explicitly discusses the system's vulnerability to adaptive attacks and data dependencies, as requested by Reviewer 6sWM.
>
> 3. **Clarified the Secondary Analyzer**: Enhanced explanation in Section 3.2 with ablation results demonstrating its critical role (9.9 F1 point drop without it).
>
> 4. **Formatting improvements**: Implemented all suggestions from Reviewer haTS (unified references, consistent heatmap colors, Algorithm 1 clarification).
>
> All new experimental results strengthen our claims without altering the core contributions.
>
> Thank you again for your constructive feedback throughout the review process.
>
> Sincerely,
> The Authors

---

> > ### Comment · Action_Editor_vUbe · 2025-10-12
> >
> > Thank you for the camera ready version, which addressed the suggested minor revision. The camera ready version is approved.
> >
> >
> > Regards,
> >
> > AE